# Development and validation of the Health Segment Classification of Population Encompassed within Singapore (HealthSCOPES) framework

Ian Yi Han Ang [1]*, Nabilah Rahman [1,2], Shing Hei Wong[3], Sheryl Hui-Xian Ng[4], Kyle Xin Quan Tan[1,5], Ke Xin Eh[3], Zheng Jye Ling [1,3,6], Andrea Su En Lim[3], Kelvin Bryan Tan[7], Sue Anne Toh[3,5,6]

1 Saw Swee Hock School of Public Health, National University of Singapore, Singapore, Singapore, 2 Biostatistics, Singapore Clinical Research Institute, Consortium for Clinical Research and Innovation, Singapore, Singapore, 3 Regional Health System Office, National University Health System, Singapore, Singapore, 4 Health Services and Outcomes Research, National Healthcare Group, Singapore, Singapore, 5 NOVI Health, Singapore, Singapore, 6 Yong Loo Lin School of Medicine, National University of Singapore, Singapore, Singapore, 7 Ministry of Health, Singapore, Singapore

* yha2103@columbia.edu

**Data Availability Statement:** Data cannot be shared publicly because there are legal restrictions on sharing the de-identified national administrative

## Abstract

### Introduction

The population is heterogeneous with varying levels of healthcare needs. Clustering individuals into health segments with more homogeneous healthcare needs allows for better understanding and monitoring of health profiles in the population, which can support data-driven resource allocation.

### Methods

Using the developed criteria, data from several of Singapore's national administrative datasets were used to classify individuals into the various health segments. Cross-sectional analysis of healthcare utilization charges was conducted. Validation was done for the framework's prognostic ability of clinically relevant outcomes measured in the following year.

### Results

The framework is comprised of twelve segments classed within four broad groups. The segments comprising individuals with cancer, with transitional care needs, and in the last year of their lives had the highest mean per resident healthcare charges. The segments comprising adults and seniors with complex chronic conditions and with transitional care needs had the highest percentage of individuals historically diagnosed with obesity. The framework was able to distinguish varying tiers of healthcare utilization charges and relative risk of death in the following year.

data consolidated and owned by the Government of Singapore. Access to the data can be requested through the TRUST Platform with approvals from the Data Access Committee (https://trustplatform.sg/collaborate/access-trust-data/).

**Funding:** The author(s) received no specific funding for this work.

**Competing interests:** NO authors have competing interests.

## Discussion

The framework was developed using a hybrid approach, with expert input and comprehensive national data that extended beyond the usual hospital patient population. The framework can be directly applied for use in program or policy design, evaluation, and cost-effectiveness analyses.

## Conclusion

The HealthSCOPES framework was developed to segment the entire population in Singapore with similar healthcare needs.

## Introduction

As populations age and prevalence of chronic diseases increase, there is a greater demand for healthcare resources and an increased burden on healthcare systems [1, 2]. Efficient utilization of finite healthcare resources is thus essential for long-term sustainability. Populations are inevitably heterogeneous, with different individuals having varying levels of healthcare needs. Clustering individuals in the population into health segments that are likely to have more homogeneous healthcare needs is a first step in better understanding and monitoring of health profile trends in the population [3–5]. This would facilitate health systems administrators and policy-makers to plan ahead with resource and infrastructure needs, and develop more targeted programs, interventions, and financial schemes that are integrated across the various settings in the care continuum [3–7].

Population health segmentation frameworks could take either a primarily quantitative data-driven approach or a qualitative expert-driven approach in their development [8]. Most of the prior framework development work has taken a hybrid approach of combining both data-driven and expert-driven approaches. With increased adoption of centralized electronic health records, and availability of hospital administrative data and insurance claims data, the use of the data-driven approach has become increasingly popular in health segmentation work. Previous models developed have however been limited to specific populations, such as elderly [9–11], or specific high-needs or high-hospital utilizer patients [12–14], which are not generalizable to the full population in a nation.

Using only hospital utilization data for population health segmentation is not person-centered, particularly since it ignores other social, economic, psychological, and behavioral determinants that influence health and that also better inform healthcare needs. Hospital utilization data is also restricted by the current healthcare structures and services provided, with most centered around hospitals without inclusion of healthcare provided outside of the hospital setting. Homogeneity achieved in hospital utilization-based clusters therefore would not necessarily allow translation of the model to anticipate and plan for future needs of the greater population. Specifically, the remaining segments of the population that have not come to contact with the hospitals or the healthcare system might be wrongly classed as healthy because there is insufficient information available. In planning ahead for unmet needs and disease prevention work, further segmentation of risk is required of those usually considered generally healthy and that have yet to have significant healthcare utilization.

An expert-driven approach to the development of population health segmentation framework often involves a workgroup of relevant personnel. This could include clinicians,

healthcare administrators, public health experts, and/or policy makers with domain knowledge in healthcare systems, public health, and healthcare policy planning [5, 15–17]. The workgroup would develop and refine the framework based on their experience and research of published theoretical and experimental works. Frameworks by Lynn et al. (2007), the Ministry of British Columbia, Canada [5], the North West London Whole Systems Integrated Care [17], and by Low et al. (2017) can be considered to primarily use an expert-driven approach. The framework by Lynn et al. (2007) has eight segments, while that by the Ministry of British Columbia, Canada [5] has 14 segments arranged into 4 broad groups. The framework by North West London Whole Systems Integrated Care [17] used a matrix form to arrange 10 segments across a dimension of three age groups. The framework by Low et al. (2017) had further split one of the five segments outlined by Singapore's Ministry of Health (MOH), generating an extension with six segments.

As part of the efforts to better understand population health in Singapore, the Health Segment Classification of Population Encompassed within Singapore (HealthSCOPES) framework was developed to segment the national population. This paper will describe the development of the framework from conceptualization through refinement to the finalized framework and the practical criteria to classify individuals into segments. This paper will also present the descriptive and the utilization charges across different key healthcare settings for each segment. Unlike other previous efforts that have focused on only a cross-sectional snapshot of the population, the objective of this paper is also to investigate the prognostic ability of the framework for relevant outcomes in the following year.

## Material and methods

### Study design

In the development and refinement of the HealthSCOPES framework, an iterative process with literature reviews, expert consultations, and data analyses was used to develop and refine the framework and its segment classification criteria. Data used was from a combination of several national administrative datasets with individual-level socio-demographic, mortality, morbidities, and healthcare utilization information. The data was used to classify individuals into the various segments based on the developed criteria. Cross-sectional analysis of healthcare utilization charges was conducted for 2016 and 2017 calendar years. This study obtained exemption approval from National Healthcare Group Domain Specific Review Board (reference number: 2018/00892) and there was no need to obtain consent as it was research that involved analysis of dataset without identifiers.

### Setting

Singapore is a city-state nation located in Southeast Asia with a total population of 5.6 million, of which over 3.9 million are citizens and permanent residents residing in the nation within the last 12 months [18]. The population has an ethnicity composition of 74% Chinese, 13% Malays, 9% Indians, and 3% Others [19], and a life expectancy of 82.9 years [18]. There are three public integrated healthcare systems in Singapore, each with a geographical region that they are tasked by the government to oversee: Central, Western, and Eastern.

### Database

National data from several administrative datasets held in MOH that covered the period from 1990 to 2017 was used. The datasets contained individual-level socio-demographic information, mortality, chronic disease diagnoses, as well as past healthcare utilization, claims, and

referrals. The data held in MOH were de-identified but linked within and between datasets using a uniquely generated identification code. The datasets were accessed in years 2018 to 2022, and the authors did not have any information that could identify individual participants during or after data collection. In total, the datasets contained information on 10,243,409 unique individuals (alive and deceased). Of which, 6,862,023 were Singaporeans or Permanent Residents.

The datasets contained healthcare utilization data submitted through the Central Claims Processing System and data submitted for subvention purposes from (i) public sector hospitals for inpatient episodes, emergency department (ED) attendances, and specialist outpatient clinic (SOC) attendances, (ii) public sector primary care centers (polyclinics), and (iii) private sector Community Health Assist Scheme (CHAS) clinics. CHAS is a scheme introduced in 2014 aimed to make primary healthcare more accessible and affordable to lower- to middle-income Singaporean households by providing higher subsidies to these households. Before November 2019, CHAS benefits were tiered depending on household income with CHAS Blue cardholder receiving higher subsidies than CHAS Orange cardholder. From November 2019, CHAS Green was introduced for Singaporean with chronic conditions who do not qualify for CHAS Blue and Orange.

The public acute hospitals and polyclinics, which handle about 80% and 20% respectively of all the patient load in Singapore [20, 21]. The data contained diagnosis codes and visit specialties, which allowed for classifying individuals with illnesses such as mental health conditions, cancer, and cognitive decline. The national chronic disease registry, Chronic Disease Management Programme (CDMP), also contained information of patients diagnosed with a list of twenty chronic diseases and managed in hospitals, polyclinics, and/or participating private primary care General Practitioner (GP) clinics. Historical diagnosis of obesity was captured in this registry, with obesity set as having body mass index of 27.5 kg/m$^2$ or higher, the criteria for Asian populations [22].

The datasets also contained claims and referrals for some nationwide standardized services and programs that had specific eligibility criteria. These criteria reflected the health and social needs of individuals who were referred to and/or enrolled in them. For instance, the eligibility criteria of admission to nursing home included being physically or mentally disabled due to illness, being semi-mobile, and/or requiring assistance in activities of daily living such as using the toilet. Being admitted to nursing homes was therefore used as a proxy to identify those who were frail and that required long-term care. A program with nationally standardized eligibility criteria was the Hospital2Home (H2H) program, which was a post-discharge transition care program. A list of potentially high healthcare utilizers generated from an artificial intelligence predictive model by the national technology agency, Synapxe (previously named Integrated Health Information Systems), was used to enroll patients into the H2H program.

The datasets also contained socio-demographic information, which included age, gender, ethnicity, housing type, as well as financial assistance schemes that an individual qualified for. These data allowed for identification of factors [23] commonly associated of having high risk of chronic diseases and increased utilization of healthcare services, such as housing type [24–28], living alone [29, 30], and requiring financial assistance [26].

## Outcome measures

Outcome measures of interest are utilization charges (unsubsidized full gross bill amounts) and utilization frequencies across different healthcare settings in Singapore. These measures of interest covered settings for acute hospital inpatient admissions, day surgery inpatient admissions, emergency department (ED) visits, and specialist outpatient clinic (SOC) visits, as well

as polyclinic visits. For inpatient admissions, the number of inpatient bed days was also included.

The HealthSCOPES segmentation criteria was applied to Singapore residents to derive 2016 broad groups and segments. Only segment criteria with available data in MOH administrative dataset at point of analysis were applied. The 2016 broad groups and segments were described in terms of demographic information and healthcare utilization. The categorical variables were summarized by frequency and percent and, continuous variables were summarized by descriptive statistics.

## Framework development

The framework was first conceptualized using the broad segment groups outlined by the Singapore MOH [16] as a basis: mostly healthy, serious acute illness but curable, stable chronic, complex chronic, and end-of-life. Using these five broad groups as a reference, four broad groups were generated titled using the care goals in mind for each of the groups: "Staying Healthy", "Living Well with Illness", "Getting Well", and "Maximizing Quality Life Years". These names for the broad groups were also designed to reframe and shift the messaging away from being negative and disease-centric to one that is positive and salutogenic. The matrix dimension of age [15, 17] was added, with those aged 0–18 years old grouped as children (and adolescents), 19–64 years old grouped as adults, 65 years old and above grouped as seniors.

Existing expert-driven frameworks that were published in peer-reviewed journals or as reports publicly available online were reviewed. Segments from expert-driven frameworks by Lynn et al. (2007) [3], the Ministry of British Columbia, Canada [5], and the North West London [17] were combined and collapsed, and a segment of "at risk of developing a chronic condition" from Langton et al. (2016) [31] added. Based on the approach recommended by Lynn et al.'s "Bridges to Health" model [3], the number and definitions for each segment were intentionally kept limited and simple with the segments include every individual, and within each included people with sufficiently similar healthcare needs and priorities. As also highlighted by the London Health Commission [15], the population health segments generated should also be stable over time, be applied in the real-world based on good judgment, and allow for budgets to be set for the entire segment.

An initial framework was developed and consultations with five local clinical experts were also conducted to get their input. The clinical experts were at the consultant level and above, and covered specialties of oncology, palliative care, pediatrics, endocrinology, and preventive medicine. The expert inputs from the consultations were to ensure that the framework was best contextualized to Singapore's population and facilitated development of practical care goals, actionable care programs, and measurable outcomes for each segment. As each clinical expert focused on segments relevant to their specialty, there were no consensus procedures taken. The classification criteria for the different segments were then developed and refined based on available data in the national-level healthcare and social administrative datasets and expert inputs. A final framework was developed with 12 segments, [A] to [L], in the four broad segment groups and across the dimension of three age groups (**Fig 1**).

Segments [A] to [C] are the three segments that make up the "Staying Healthy" broad group, who have no major health needs. Segment [A] comprises individuals of all ages that are mostly healthy and with no known chronic conditions and no known clinical and social risk factors. Segments [B] and [C] comprise children and adults, and seniors, respectively, that are considered to be at risk. These individuals all have no known chronic conditions but do have at least one clinical or social risk factor (such as low socio-economic status or obesity).

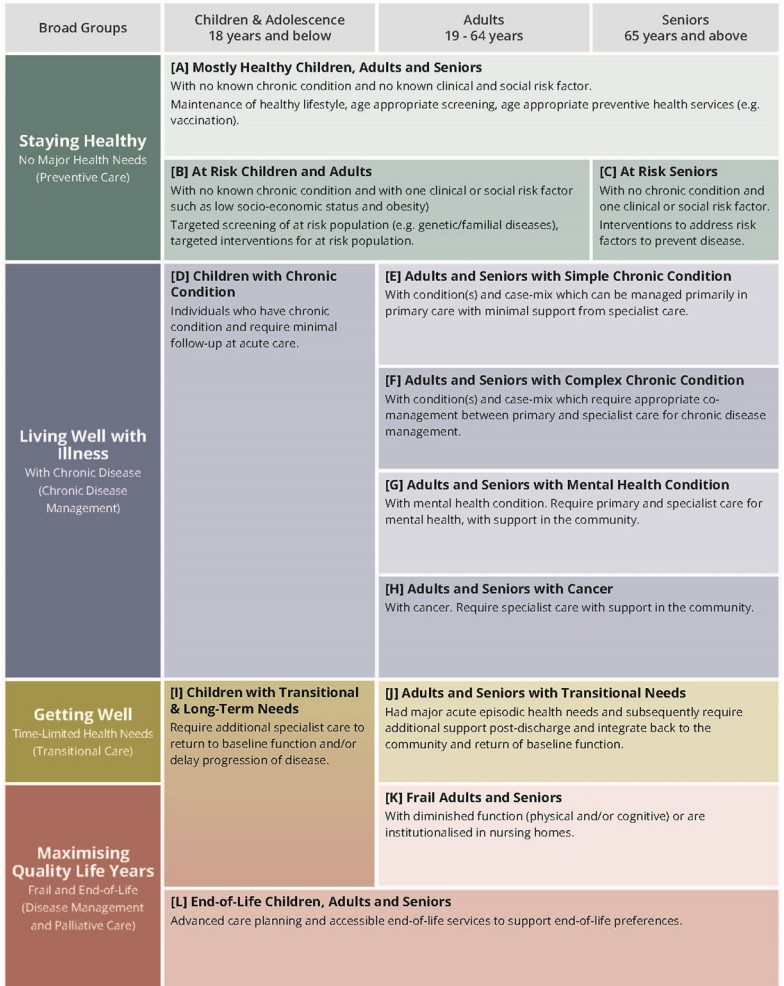

**Fig 1. The final Health Segment Classification of Population Encompassed within Singapore (HealthSCOPES) framework with 12 segments in four broad segment groups and across the dimension of three age groups.**

Segments [D] to [H] are the five segments that make up the "Living Well with Illness" broad group, who have been diagnosed in the system with at least one chronic illness. Segment [D] comprises children with known chronic conditions that are fairly stable and that require minimal follow-up at acute care settings. Segment [E] includes adults and seniors with simple chronic conditions that can be managed primarily in primary care with minimal support from specialist care. Segment [F] comprises adults and seniors with complex chronic conditions that require some management by specialist care. Segment [G] comprises adults and seniors with mental health conditions, while segment [H] includes adults and seniors with cancer.

Patients with transient, acute healthcare needs fall into Segment [I] and [J]. These individuals would have transitional care needs that help transition their care from hospital to community or home setting to aid in their continued recovery to baseline function. Segment [I] comprises children that have transitional and long-term care needs and that require additional specialist care to return baseline functions and/or to delay the progression of their diseases. Segment [J] comprises adults and seniors that have experienced a major acute health episode and thus require additional support after discharge from hospital.

Segments [K] and [L], together Segment [I], make up the "Maximizing Quality Life Years" broad group. Segment [K] comprises adults and seniors that have diminished physical and/or cognitive functions and are considered frail, and/or that are living in a long-term care institution such as a nursing home. Segment [L] comprises individuals of all ages that are in the final year of their lives or were referred to palliative care services, and so require advanced care planning and services that support their end-of-life preferences.

## Classification criteria

Together, all the segments covered the entire population and were tiered from highest to lowest healthcare needs (Fig 2). At any point, individuals would fall into only one of the segments based on the definition. As their health state changed across time points, individuals could be reclassified into another segment, but never in more than one segment at a time. The classification of individuals was based on meeting criteria of a segment with the highest healthcare needs in one calendar year, and that overrode classification into a segment with lower healthcare needs. The developed hierarchy of classification is presented visually as a flowchart as shown in Fig 2.

## Statistical analyses

Descriptive analyses of demographics were performed for each segment. Yearly healthcare utilization charges were calculated for each of the different settings for the 2016 calendar year. Total yearly healthcare utilization charges were calculated from summing yearly healthcare utilization charges across the various healthcare settings and presented as a total dollar amount incurred for each segment (total segment healthcare charges) and as a mean dollar amount per person of the segment (mean per resident healthcare charges).

In order to validate the HealthSCOPES framework, the prognostic ability of the 2016 broad groups and segments, on clinically relevant outcomes measured in 2017, were studied. Prognostic ability of segments was used as a proxy indicator of clinically meaningful, needs-based segmentation. Differing expected healthcare utilization and mortality among the different segments shows that the segmentation framework could be a potential tool to facilitate planning of healthcare resources and interventions. Only residents with assigned broad groups and segments in 2016 who continued to live in 2017 were included in the validation. The outcomes studied were (i) number of polyclinic visits, (ii) number of SOC visits, (iii) number of ED visits, (iv) number of inpatient admissions, (v) total inpatient bed days (BD), (vi) mortality, and (vii) admission to intensive care unit (ICU).

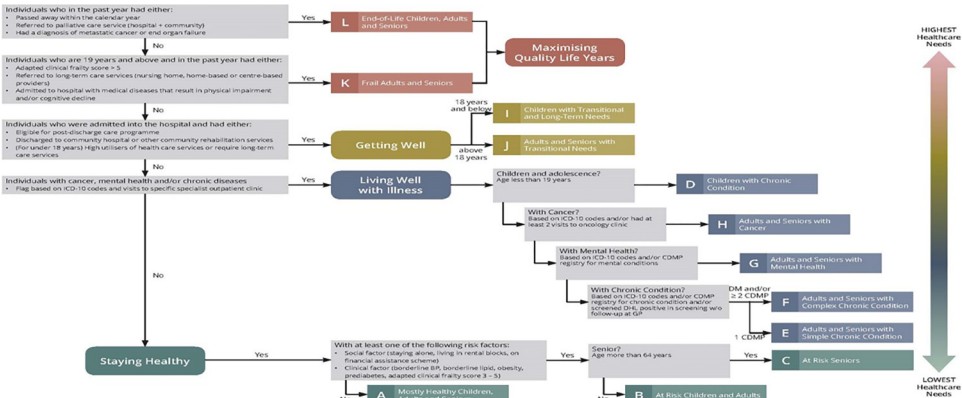

**Fig 2. The classification hierarchy outlining the criteria to categorize an individual into one of the 12 health segments.**

Two-part models (twopm) with offset term and broad groups or segments as independent variable were used to model dependent variables (i)-(v) in order to calculate the marginal effects (ME). Twopm was used as a large fraction of the residents did not have any healthcare utilization. Twopm consists of two parts. The first part is a binary component which models the probability of using the specific healthcare service. The second part is a continuous component which models non-zero healthcare utilization using generalized linear model (GLM). Logit-link and log-link were used for the first part and second part of the twopm respectively. Offset term was used to account for patients who died in 2017 and hence less opportunity for utilization. For each dependent variable, Poisson, Gamma, Negative Binomial, and Gaussian family were considered for the second part of the twopm. The distribution family with the lower Akaike information criterion (AIC) was selected and the results reported. Marginal effects can be interpreted as the average change in 2017 utilization when there was a change (hypothetical) in broad group or segment in 2016 from reference.

With broad groups or segments as independent variable, GLM with logit link and binomial family was used to model presence of ICU admission in 2017 and to generate odds ratio (OR). Mortality was modelled using Cox-proportional hazard models to derive hazard ratio with broad groups or segments as independent variable without any other covariate.

For both twopm and GLM, the group or segment with the least healthcare needs was used as the reference group (i.e. "Staying Healthy" group and [A] Mostly Healthy Children, Adults & Seniors segment). As deaths are expectedly rare in the "Staying Healthy" group and [A] Mostly Healthy Children, Adults & Seniors segment, the "Living Well with Illness" group and [E] Adults & Seniors with Simple Chronic Condition segment were used as reference group in the survival analyses for a more informative comparison. For the broad group analysis, [I] Children with Transitional & Long-Term Needs was classed only within the "Getting Well" group and not in the "Maximizing Quality Life Years". Moreover, the segments encompassing solely children were dropped in the segment-level survival analysis as deaths were rare in those segments. There were 13 deaths in [D] Children with Chronic Condition and 24 deaths in [I] Children with Transitional & Long-Term Needs.

Statistical significance was assessed using a threshold of 0.05. Point estimates and 95% confidence intervals (CI) of the marginal effects, odds ratios and hazard ratios will be reported. STATA version 14.1 (Stata Corp, College Station, Texas, USA) was used to segment the population and perform twopm. R Version 3.6 was used to perform the logistic regression and survival analyses. The R package 'survival' was specifically used to perform the latter analyses. Forest plots found in this paper were produced using the R package 'forestplot'.

## Results

### Demographics

Using the database master list of individuals, exclusion criteria were applied to form 2016 base population for segmentation (**Fig 3**). Those without race information were excluded. To exclude individuals who might be missing death date, we excluded those aged above 110 in 2016. There was a total of 4,671,465 residents that were alive at any point in 2016 (**Table 1**). The size of each of the broad groups and segments followed the order of healthcare needs–the size decreased with increasing healthcare needs. The majority of the population was part of the "Staying Healthy" group (n = 3,489,881, percentage of total = 74.7%). The sizes of the two other broad groups (with segment [I] classed within the "Getting Well") and 12 segments can be found in **Table 2**. Based on percentages of the total in each geographical region, there was no significant difference in distribution of broad groups and segments across the different regions, indicating homogeneity within Singapore.

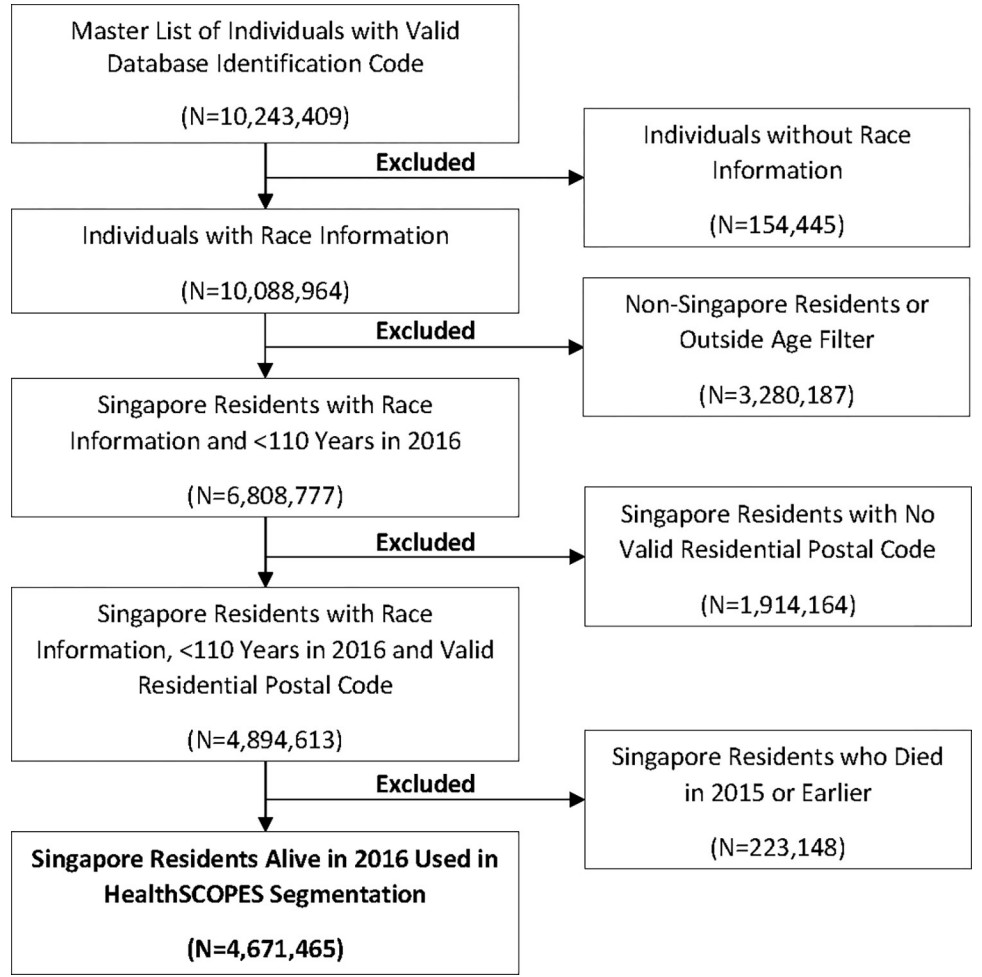

**Fig 3. Flow chart of exclusion criteria to obtain analysis population.**

Table 2 also provides summary statistics on demographic information with no missing data. Examining only the segments including adults, segment [K] had the highest mean age followed by segments [J], [F], [H], [E], and [G]. There was higher percentage of males in all segments except for segments [E], [G], [H], [J], and [K]. Compared to segment [A], there was higher percentage (more than 5%) of ethnic Chinese in segments [C], [E], [F], [G], [H], [K], and [L]. Compared to segment [A], there was also a higher percentage of ethnic Malay in the segments [B], [D], [I], [J], and [L]. Although requiring financial assistance is not a classification criterion for segments [J], [K], and [L], more than 40% of the residents in the segments were beneficiaries of Community Health Assist Scheme (CHAS). Similarly, segments [F] and [J] had the highest percentage of residents historically diagnosed with obesity even though historical diagnosis of obesity is not a classification criterion. The other patterns in demographics statistics were an artefact of the framework classification criteria.

## Healthcare utilization

The total segment healthcare charges (with breakdown of yearly healthcare utilization charges across different settings) and the mean per resident healthcare charges are shown in **Fig 4** with no further missing data. Segment [A] had the highest total segment healthcare utilization

**Table 1. Size of the 4 broad groups and 12 segments by regions.**

| | Central | | Western | | Eastern | | Total | |
|---|---|---|---|---|---|---|---|---|
| **Number of Individuals** | 1,845,900 | | 1,313,618 | | 1,511,947 | | 4,671,465 | |
| **Staying Healthy, n (%)** | 1,355,422 | (73.4) | 990,306 | (75.4) | 1,144,153 | (75.7) | 3,489,881 | (74.7) |
| [A] Mostly Healthy Children, Adults & Seniors | 1,141,083 | (61.8) | 853,950 | (65.0) | 981,316 | (64.9) | 2,976,349 | (63.7) |
| [B] At Risk Children & Adults | 198,404 | (10.8) | 127,808 | (9.7) | 152,557 | (10.1) | 478,769 | (10.3) |
| [C] At Risk Seniors | 15,935 | (0.9) | 8,548 | (0.7) | 10,280 | (0.7) | 34,763 | (0.7) |
| **Living Well with Illness, n (%)** | 446,024 | (24.2) | 296,493 | (22.6) | 336,142 | (22.2) | 1,078,659 | (23.1) |
| [D] Children with Chronic Condition | 6,127 | (0.3) | 4,789 | (0.4) | 6,290 | (0.4) | 17,206 | (0.4) |
| [E] Adults and Seniors with Simple Chronic Condition | 140,747 | (7.6) | 98,036 | (7.5) | 110,150 | (7.3) | 348,933 | (7.5) |
| [F] Adults and Seniors with Complex Chronic Condition | 239,973 | (13.0) | 155,292 | (11.8) | 177,701 | (11.8) | 572,966 | (12.3) |
| [G] Adults and Seniors with Mental Health Condition | 49,764 | (2.7) | 30,517 | (2.3) | 34,354 | (2.3) | 114,635 | (2.5) |
| [H] Adults and Seniors with Cancer | 9,413 | (0.5) | 7,859 | (0.6) | 7,647 | (0.5) | 24,919 | (0.5) |
| **Getting Well, n (%)** | 12,313 | (0.7) | 9,243 | (0.7) | 9,913 | (0.7) | 31,469 | (0.7) |
| [I] Children with Transitional & Long-Term Needs | 2,008 | (0.1) | 1,431 | (0.1) | 1,792 | (0.1) | 5,231 | (0.1) |
| [J] Adults and Seniors with Transitional Needs | 10,305 | (0.6) | 7,812 | (0.6) | 8,121 | (0.5) | 26,238 | (0.6) |
| **Maximizing Quality Life Years, n (%)** | 32,141 | (1.7) | 17,576 | (1.3) | 21,739 | (1.4) | 71,456 | (1.5) |
| [K] Frail Adults and Seniors | 17,238 | (0.9) | 9,212 | (0.7) | 11,093 | (0.7) | 37,543 | (0.8) |
| [L] End-of-Life Children, Adults and Seniors | 14,903 | (0.8) | 8,364 | (0.6) | 10,646 | (0.7) | 33,913 | (0.7) |

charges standing at about $1 billion but the lowest mean per resident healthcare charges. The mean healthcare charges per resident were the highest for segment [J], followed by segment [L]. As a result, even though the population size of segments [J] and [L] were about 1% that of segment [A], these segments respectively incurred about 31% and 40% of segment [A]'s total healthcare charges. Although the mean per resident healthcare charges for segments [H]-[L] were about 4–9 times higher than that of segment [F], the population size of segment [F] was about 20-fold the size of segments [H] and [L]. This led to the total segment healthcare charges of [F] to be the second highest at close to $790 million, and over 2–18 times that of segments [H]-[L].

Looking at the contribution of different settings to the total segment healthcare charges, a sizeable proportion of the charges for segments [E]-[H] could be attributed to the inpatient admissions, SOC visits, and day surgery admissions. There was also a sizable contribution of healthcare utilization charges from polyclinic visits in segment [F]. In contrast, for segments [J]-[L], the bulk of the charges were driven by inpatient admissions, with some proportion incurred at the SOCs, while there was minimal relative contribution from polyclinics and day surgery.

## Validation

Of the 4,701,074 residents segmented for 2016, 4,653,020 were alive in 2017. These 4,653,020 residents were included in the validation analyses. There was no further missing data.

**Broad group validation.** Gamma was selected as the family for the second part of the twopm modeling number of SOC visits (AIC = 4.56), total inpatient bed days (AIC = 5.90) and number of ED visits (AIC = 2.77). Poisson was selected as the family for the second part of the twopm modeling number of polyclinic visits (AIC = 4.24) and number of inpatient admissions (AIC = 2.77).

The estimated marginal effects and odds ratios showed that there was variation in 2017 healthcare utilization amongst the broad groups segmented for 2016 (Table 3). With the "Staying Healthy" broad group as reference, the marginal effects were all larger than 0 and the odds

**Table 2. Size and demographics of the 12 different segments.**

| | [A] | [B] | [C] | [D] | [E] | [F] | [G] | [H] | [I] | [J] | [K] | [L] |
|---|---|---|---|---|---|---|---|---|---|---|---|---|
| **Number of Individuals (%)** | 2,976,349 | 478,769 | 34,763 | 17,206 | 348,933 | 572,966 | 114,635 | 24,919 | 5,231 | 26,238 | 37,543 | 33,913 |
| **Age, Mean (SD)** | 29.7 (18.0) | 29.3 (18.4) | 71.9 (8.0) | 7.2 (6.2) | 49.2 (15.8) | 63.4 (12.5) | 48.5 (16.8) | 60.1 (13.5) | 7.5 (6.2) | 63.5 (17.0) | 76.5 (12.4) | 71.1 (15.5) |
| **Gender, n (%)** | | | | | | | | | | | | |
| Male | 1,510,131 (50.7) | 244,453 (51.1) | 17,571 (50.6) | 9,597 (55.8) | 171,405 (49.1) | 290,491 (50.7) | 48,273 (42.1) | 9,110 (36.6) | 2,891 (55.3) | 12,691 (48.4) | 15,973 (42.6) | 18,404 (54.3) |
| Female | 1,466,218 (49.3) | 234,316 (48.9) | 17,192 (49.5) | 7,609 (44.2) | 177,528 (50.9) | 282,475 (49.3) | 66,362 (57.9) | 15,809 (63.4) | 2,340 (44.7) | 13,547 (51.6) | 21,570 (57.5) | 15,509 (45.7) |
| **Ethnicity, n (%)** | | | | | | | | | | | | |
| Chinese | 1,954,913 (65.7) | 287,716 (60.1) | 28,340 (81.5) | 10,273 (59.7) | 249,267 (71.4) | 426,095 (74.4) | 86,274 (75.3) | 20,586 (82.6) | 3,135 (59.9) | 17,618 (67.2) | 30,863 (82.2) | 24,996 (73.7) |
| Malay | 226,674 (7.6) | 98,770 (20.6) | 2,237 (6.4) | 3,073 (17.9) | 38,251 (11.0) | 57,158 (10.0) | 9,236 (8.1) | 1,538 (6.2) | 897 (17.2) | 3,643 (13.9) | 2,626 (7.0) | 4,444 (13.1) |
| Indian | 240,652 (8.1) | 34,837 (7.3) | 2,368 (6.8) | 1,767 (10.3) | 27,676 (7.9) | 48,078 (8.4) | 10,111 (8.8) | 1,236 (5.0) | 585 (11.2) | 2,690 (10.3) | 2,175 (5.8) | 2,683 (7.9) |
| Others | 554,110 (18.6) | 57,446 (12.0) | 1,818 (5.2) | 2,093 (12.2) | 33,739 (9.7) | 41,635 (7.3) | 9,014 (7.9) | 1,559 (6.3) | 614 (11.7) | 2,287 (8.7) | 1,879 (5.0) | 1,790 (5.3) |
| **Housing, n (%)** | | | | | | | | | | | | |
| Rental (public) | 0 (0.0) | 53,180 (11.1) | 7,497 (21.6) | 665 (3.9) | 11,055 (3.2) | 28,600 (5.0) | 5,568 (4.9) | 763 (3.1) | 234 (4.5) | 2,055 (7.8) | 3,728 (9.9) | 1,917 (5.7) |
| Non-rental (public) | 2,295,354 (77.1) | 413,862 (86.4) | 26,225 (75.4) | 14,034 (81.6) | 283,838 (81.3) | 476,069 (83.1) | 94,704 (82.6) | 19,283 (77.4) | 4,280 (81.8) | 21,922 (83.6) | 29,488 (78.5) | 28,491 (84.0) |
| Private housing | 680,995 (22.9) | 11,727 (2.5) | 1,041 (3.0) | 2,507 (14.6) | 54,040 (15.5) | 68,297 (11.9) | 14,363 (12.5) | 4,873 (19.6) | 717 (13.7) | 2,261 (8.6) | 4,327 (11.5) | 3,505 (10.3) |
| **Financial Assistance, n (%)** | | | | | | | | | | | | |
| Non-CHAS | 2,680,631 (90.1) | 76,009 (15.9) | 12,609 (36.3) | 11,636 (67.6) | 208,204 (59.7) | 276,620 (48.3) | 61,165 (53.4) | 14,031 (56.3) | 3,245 (62.0) | 10,955 (41.8) | 15,272 (40.7) | 14,642 (43.2) |
| CHAS—Orange | 295,718 (9.9) | 18,952 (4.0) | 2,335 (6.7) | 1,955 (11.4) | 58,047 (16.6) | 100,409 (17.5) | 17,312 (15.1) | 3,675 (14.8) | 679 (13.0) | 4,279 (16.3) | 4,380 (11.7) | 4,611 (13.6) |
| CHAS—Blue | 0 (0.0) | 383,808 (80.2) | 19,819 (57.0) | 3,615 (21.0) | 82,682 (23.7) | 195,937 (34.2) | 36,158 (31.5) | 7,213 (29.0) | 1,307 (25.0) | 11,004 (41.9) | 17,891 (47.7) | 14,660 (43.2) |
| **Special Financial Scheme, n (%)** | | | | | | | | | | | | |
| Pioneer Generation | 30,061 (1.0) | 0 (0.0) | 23,295 (67.0) | 0 (0.0) | 39,789 (11.4) | 220,602 (38.5) | 16,241 (14.2) | 7,813 (31.4) | 0 (0.0) | 12,068 (46.0) | 29,236 (77.9) | 20,801 (61.3) |
| **Historically diagnosed with Obesity, n (%)** | 0 (0.0) | 11,927 (2.5) | 160 (0.5) | 265 (1.5) | 9,573 (2.7) | 66,643 (11.6) | 6,190 (5.4) | 1,436 (5.8) | 147 (2.8) | 2,877 (11.0) | 2,221 (5.9) | 2,934 (8.7) |

ratios were all larger than 1. Larger variation was observed in SOC visits (ME: 1.80 to 4.40), total inpatient bed days (ME: 0.98–8.88) and ICU admission (OR: 6.59 to 27.84).

The "Getting Well" group had the highest number of polyclinic visits, SOC visits, inpatient admissions and ED visits when compared to the reference group of "Staying Healthy". For instance, the "Getting Well" group on average utilized 6.05 (95% CI: 5.97–6.14) more SOC visits than the "Staying Healthy" group.

The "Maximizing Quality Life Years" group had the highest total inpatient bed days in the following year. The group on average utilized 8.88 (95% CI: 8.66–9.11) more bed days in the following year than the "Staying Healthy" group. The odds of ICU admission in 2017 for the "Maximizing Quality Life Years" group was the highest, standing at 27.84 times (95% CI: 26.35–29.40) that of the "Staying Healthy" group.

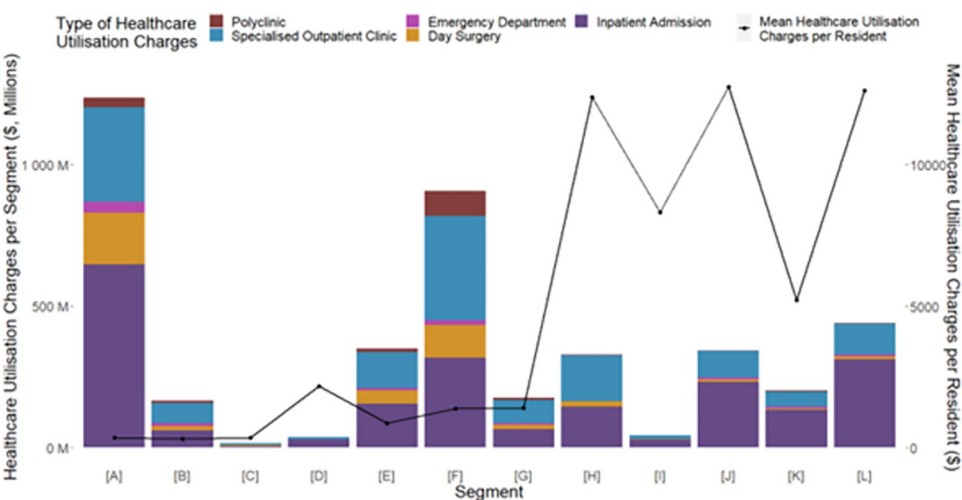

**Fig 4. Total healthcare utilization charges across different care settings and the mean total healthcare utilization charges per resident.**

**Table 3. Marginal effects and odd ratios relating to 2017 healthcare utilization of 2016 HealthSCOPES broad groups residents.**

| Variable | Broad Group | n | ME | 95% CI | P-value |
|---|---|---|---|---|---|
| No. of polyclinic visits | [A]-[C] | 3,489,877 | Reference | | |
| | [D]-[H] | 1,078,655 | 2.04 | 2.03–2.04 | < .001 |
| | [I]-[J] | 31,469 | 2.21 | 2.17–2.24 | < .001 |
| | [K]-[L] | 53,019 | 2.12 | 2.09–2.14 | < .001 |
| No. of specialist outpatient clinic visits | [A]-[C] | 3,489,877 | Reference | | |
| | [D]-[H] | 1,078,655 | 1.80 | 1.79–1.81 | < .001 |
| | [I]-[J] | 31,469 | 6.05 | 5.97–6.14 | < .001 |
| | [K]-[L] | 53,019 | 4.40 | 4.35–4.46 | < .001 |
| No. inpatient admissions | [A]-[C] | 3,489,877 | Reference | | |
| | [D]-[H] | 1,078,655 | 0.15 | 0.15–0.15 | < .001 |
| | [I]-[J] | 31,469 | 1.16 | 1.14–1.18 | < .001 |
| | [K]-[L] | 53,019 | 1.07 | 1.05–1.08 | < .001 |
| Total inpatient bed days | [A]-[C] | 3,489,877 | Reference | | |
| | [D]-[H] | 1,078,655 | 0.98 | 0.97–1.00 | < .001 |
| | [I]-[J] | 31,469 | 7.14 | 6.90–7.38 | < .001 |
| | [K]-[L] | 53,019 | 8.88 | 8.66–9.11 | < .001 |
| No. of emergency department visits | [A]-[C] | 3,489,877 | Reference | | |
| | [D]-[H] | 1,078,655 | 0.18 | 0.17–0.18 | < .001 |
| | [I]-[J] | 31,469 | 1.12 | 1.10–1.14 | < .001 |
| | [K]-[L] | 53,019 | 1.03 | 1.01–1.04 | < .001 |
| ICU admission | [A]-[C] | 3,489,877 | Reference | | |
| | [D]-[H] | 1,078,655 | 6.59 | 6.36–6.83 | < .001 |
| | [I]-[J] | 31,469 | 27.13 | 25.35–29.01 | < .001 |
| | [K]-[L] | 53,019 | 27.84 | 26.35–29.40 | < .001 |

[A]-[C]: staying healthy; [D]-[H]: living well with illness; [I]-[J]: getting well; [K]-[L]: maximizing quality life years; CI: confidence interval; ME: marginal effect; OR: odds ratio; ICU: intensive care unit.

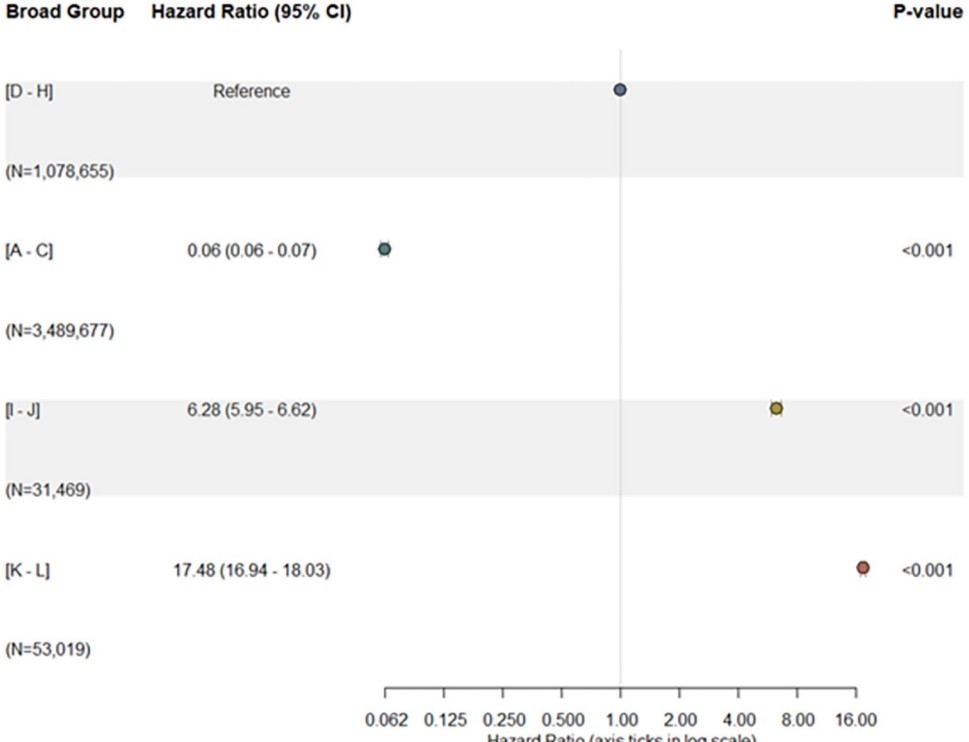

**Fig 5. Hazard ratios based on deaths in 2017 of 2016 HealthSCOPES broad groups residents.** [A]-[C]: staying healthy; [D]-[H]: living well with illness; [I]-[J]: getting well; [K]-[L]: maximizing quality life years.

Hazard ratios were higher in broad groups with higher health care needs (**Fig 5**). For instance, the death rate per unit time in the "Maximizing Quality Life Years" group was 17.48 (95% CI: 16.94–18.03) times that of the "Living Well with Illness" group.

**Segment validation.** Gamma was selected as the family for the second part of the twopm modeling number of SOC visits (AIC = 4.54), total inpatient bed days (AIC = 5.86) and number of ED visits (AIC = 2.77). Poisson was selected as the family for the second part of the twopm modeling number of polyclinic visits (AIC = 4.16) and number of inpatient admissions (AIC = 2.76). The estimated marginal effects and odds ratios showed that there was variation in 2017 healthcare utilization amongst the segments for 2016 (**Table 4**).

With segment [A] as reference, the marginal effects were all larger than 0 and the odds ratios were all larger than 1. Larger variation was observed in SOC visits (ME: 0.18 to 8.36), total inpatient bed days (ME: 0.11–11.69) and ICU admission (OR: 1.42–69.59). Differentiated MEs and ORs were also observed in segments belonging to the same broad groups. An example would be the number of SOC visits of the segments [D]-[H] in "Living Well with Illness" group, which was on average 0.91 to 8.36 higher when compared to the reference segment [A].

Segment [F] had the highest number of polyclinic visits when compared to the reference segment. On average, the segment was estimated to have visited the polyclinics 2.93 (95% CI: 2.93–2.94) more in the following year when compared to the reference segment. Segment [H] had the highest number of SOC visits, with an estimated 8.36 (95% CI: 8.24–8.49) more SOC visits when compared to the reference segment. Segment [L] had the highest inpatient admissions, total inpatient bed days, and ED visits when compared to the reference segment. For instance, the segment on average utilized 11.69 (95% CI: 11.22–12.17) more bed days in the following year than the reference segment. The odds of ICU admission in 2017 for segment

**Table 4. Marginal effects and odd ratios relating to 2017 healthcare utilization of 2016 HealthSCOPES segmented residents.**

| Variable | Segment | n | ME | 95% CI | P-value |
|---|---|---|---|---|---|
| No. of polyclinic visits | [A] | 2,976,346 | Reference | | |
| | [B] | 478,769 | 0.44 | 0.43–0.44 | < .001 |
| | [C] | 34,762 | 0.31 | 0.30–0.33 | < .001 |
| | [D] | 17,206 | 0.97 | 0.94–1.00 | < .001 |
| | [E] | 348,933 | 0.94 | 0.94–0.95 | < .001 |
| | [F] | 572,964 | 2.93 | 2.93–2.94 | < .001 |
| | [G] | 114,634 | 1.72 | 1.70–1.73 | < .001 |
| | [H] | 24,918 | 1.63 | 1.60–1.66 | < .001 |
| | [I] | 5,231 | 1.03 | 0.98–1.08 | < .001 |
| | [J] | 26,238 | 2.52 | 2.48–2.55 | < .001 |
| | [K] | 37,539 | 1.97 | 1.94–2.00 | < .001 |
| | [L] | 15,480 | 2.69 | 2.64–2.74 | < .001 |
| No. of specialist outpatient clinic visits | [A] | 2,976,346 | Reference | | |
| | [B] | 478,769 | 0.18 | 0.17–0.18 | < .001 |
| | [C] | 34,762 | 0.69 | 0.66–0.72 | < .001 |
| | [D] | 17,206 | 1.20 | 1.16–1.23 | < .001 |
| | [E] | 348,933 | 0.91 | 0.90–0.92 | < .001 |
| | [F] | 572,964 | 2.02 | 2.01–2.03 | < .001 |
| | [G] | 114,634 | 2.39 | 2.36–2.41 | < .001 |
| | [H] | 24,918 | 8.36 | 8.24–8.49 | < .001 |
| | [I] | 5,231 | 5.24 | 5.07–5.42 | < .001 |
| | [J] | 26,238 | 6.25 | 6.16–6.35 | < .001 |
| | [K] | 37,539 | 3.58 | 3.53–3.64 | < .001 |
| | [L] | 15,480 | 6.50 | 6.37–6.63 | < .001 |
| No. inpatient admissions | [A] | 2,976,346 | Reference | | |
| | [B] | 478,769 | 0.02 | 0.02–0.02 | < .001 |
| | [C] | 34,762 | 0.07 | 0.06–0.07 | < .001 |
| | [D] | 17,206 | 0.12 | 0.11–0.13 | < .001 |
| | [E] | 348,933 | 0.07 | 0.07–0.07 | < .001 |
| | [F] | 572,964 | 0.19 | 0.19–0.19 | < .001 |
| | [G] | 114,634 | 0.17 | 0.17–0.18 | < .001 |
| | [H] | 24,918 | 0.54 | 0.52–0.55 | < .001 |
| | [I] | 5,231 | 0.78 | 0.74–0.82 | < .001 |
| | [J] | 26,238 | 1.24 | 1.22–1.26 | < .001 |
| | [K] | 37,539 | 0.84 | 0.82–0.85 | < .001 |
| | [L] | 15,480 | 1.64 | 1.61–1.67 | < .001 |
| Total inpatient bed days | [A] | 2,976,346 | Reference | | |
| | [B] | 478,769 | 0.11 | 0.10–0.12 | < .001 |
| | [C] | 34,762 | 0.75 | 0.68–0.82 | < .001 |
| | [D] | 17,206 | 0.41 | 0.36–0.45 | < .001 |
| | [E] | 348,933 | 0.33 | 0.32–0.34 | < .001 |
| | [F] | 572,964 | 1.19 | 1.17–1.21 | < .001 |
| | [G] | 114,634 | 1.71 | 1.66–1.77 | < .001 |
| | [H] | 24,918 | 3.43 | 3.27–3.60 | < .001 |
| | [I] | 5,231 | 2.87 | 2.58–3.16 | < .001 |
| | [J] | 26,238 | 8.02 | 7.74–8.30 | < .001 |
| | [K] | 37,539 | 7.75 | 7.51–7.99 | < .001 |
| | [L] | 15,480 | 11.69 | 11.22–12.17 | < .001 |

(*Continued*)

**Table 4.** (Continued)

| Variable | Segment | n | ME | 95% CI | P-value |
|---|---|---|---|---|---|
| No. of emergency department visits | [A] | 2,976,346 | Reference | | |
| | [B] | 478,769 | 0.07 | 0.07–0.07 | < .001 |
| | [C] | 34,762 | 0.05 | 0.05–0.06 | < .001 |
| | [D] | 17,206 | 0.20 | 0.19–0.21 | < .001 |
| | [E] | 348,933 | 0.11 | 0.10–0.11 | < .001 |
| | [F] | 572,964 | 0.21 | 0.20–0.21 | < .001 |
| | [G] | 114,634 | 0.29 | 0.29–0.30 | < .001 |
| | [H] | 24,918 | 0.32 | 0.31–0.33 | < .001 |
| | [I] | 5,231 | 0.66 | 0.63–0.70 | < .001 |
| | [J] | 26,238 | 1.22 | 1.20–1.25 | < .001 |
| | [K] | 37,539 | 0.88 | 0.86–0.90 | < .001 |
| | [L] | 15,480 | 1.42 | 1.38–1.45 | < .001 |
| ICU admission | [A] | 2,976,346 | Reference | | |
| | [B] | 478,769 | 1.42 | 1.32–1.54 | < .001 |
| | [C] | 34,762 | 5.87 | 5.13–6.68 | < .001 |
| | [D] | 17,206 | 2.73 | 2.07–3.53 | < .001 |
| | [E] | 348,933 | 3.77 | 3.55–4.00 | < .001 |
| | [F] | 572,964 | 9.35 | 8.97–9.75 | < .001 |
| | [G] | 114,634 | 5.27 | 4.85–5.71 | < .001 |
| | [H] | 24,918 | 22.29 | 20.45–24.26 | < .001 |
| | [I] | 5,231 | 14.90 | 11.98–18.28 | < .001 |
| | [J] | 26,238 | 33.09 | 30.78–35.54 | < .001 |
| | [K] | 37,539 | 15.75 | 14.49–17.09 | < .001 |
| | [L] | 15,480 | 69.59 | 64.98–74.47 | < .001 |

[A]: mostly healthy children, adults & seniors; [B]: at risk children & adults; [C]: at risk seniors; [D]: children with chronic condition; [E]: adults & seniors with simple chronic condition; [F]: adults & seniors with complex chronic condition; [G]: adults & seniors with mental health condition; [H]: adults & seniors with cancer; [I]: children with transitional & long-term care needs; [J]: adults & seniors with transitional care needs; [K]: frail adults & seniors; [L]: end-of-life; CI: confidence interval; ME: marginal effect; OR: odds ratio; ICU: intensive care unit.

[L] were the highest, standing at 69.59 times (95% CI: 64.98–74.47) that of the reference segment.

An overall increasing trend in hazard ratios were observed in segments with higher health care needs (**Fig 6**). For instance, the death rate per unit time in segment [L] was 89.27 (95% CI: 82.77–96.28) times that of [E]. However, some exceptions were observed. With segment [E] as reference, the hazard ratio of segment [C] was larger than 1 (i.e. 3.63 (95% CI: 3.19–4.13)). The hazard ratio of segment [G] was lower than segment [F]. The hazard ratio of segment [H] was similar to that of segment [J].

## Discussion

We developed a framework to segment the Singaporean resident population into four broad groups and 12 segments, based on a set of hierarchical criteria operationalizable in administrative data. Segments comprising individuals with cancer, with transitional care needs, and in the last year of their lives had the highest mean per resident healthcare charges. These findings mirror that of a previous study using data from an academic medical center in Singapore that found cancers and inpatient deaths as being major drivers of hospital healthcare expenditures

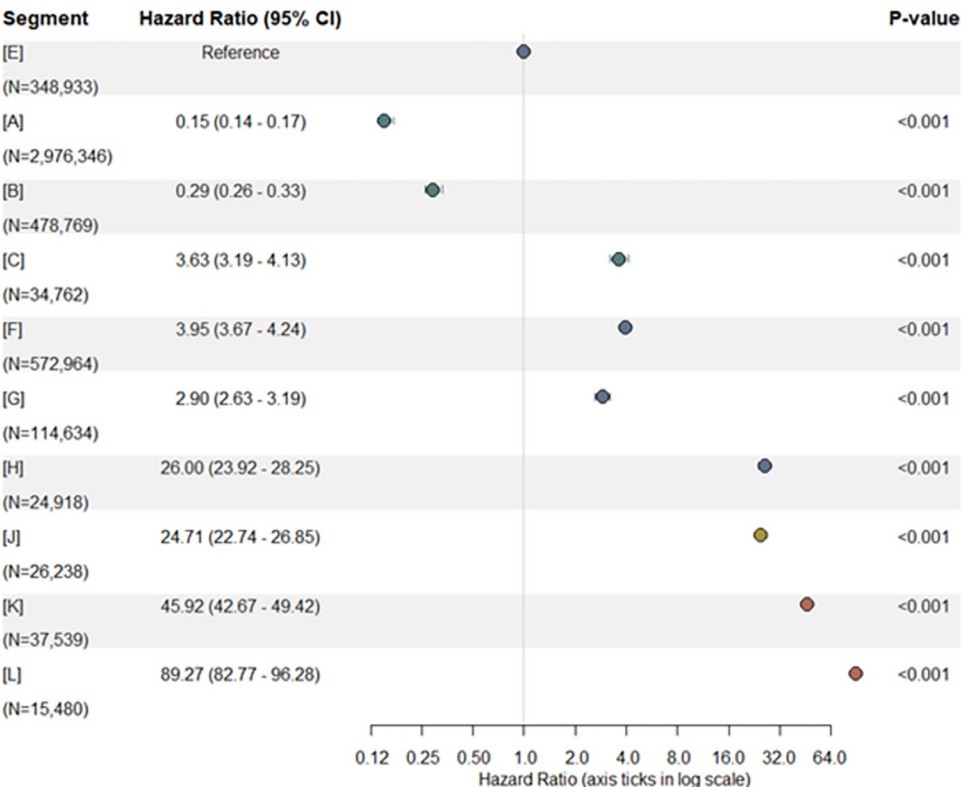

**Fig 6. Hazard ratios based on deaths in 2017 of 2016 HealthSCOPES segmented residents.** [A]: mostly healthy children, adults & seniors; [B]: at risk children & adults; [C]: at risk seniors; [E]: adults & seniors with simple chronic condition; [F]: adults & seniors with complex chronic condition; [G]: adults & seniors with mental health condition; [H]: adults & seniors with cancer; [J]: adults & seniors with transitional care needs; [K]: frail adults & seniors; [L]: end-of-life.

[32]. This indicates that when individuals move into these segments, the health utilization charges for them would be expected to escalate. At current state, these segments make up about 2% of the population. Growth in size of these segments would thus majorly add to the extended burden on the healthcare system and potentially cripple it.

Strategic policies and scalable care models need to be designed to provide continued optimal care population-wide, while also working on preventing the growth of size of these segments in the future. Existing interventions, such as post-discharge transition care programs, might help better integrate care and decrease mortality, but might not be significantly decreasing healthcare utilization charges as intended [33]. Individuals with chronic diseases are at greater risk of having acute health episodes and/or complications that incur additional healthcare resources. National efforts would therefore need to focus on sustainable long-term monitoring and management of chronic diseases in the community to ensure that these individuals can continue to live well and not worsen. Going further upstream, early detection and prevention of chronic diseases through screening and educational programs would be crucial to ensure that the generally healthy individuals continue to stay healthy.

## Strengths

One of the key strengths of this effort was the hybrid approach of being both qualitatively expert-driven and quantitatively data-driven. The consultations with local clinical experts

ensured that the framework was best contextualized to Singapore's population. The use of comprehensive national data from several administrative datasets extended beyond only the patient pool in the acute hospital setting and allowed coverage of the entire nation's population to include those that have not come to contact by hospitals or the healthcare system. However, the datasets could benefit from having additional sources of data, such as health behaviors and psychosocial factors, which would usually only be available if routinely collected systematically with using questionnaires for these patient-reported outcomes.

Another strength of this effort was the validation of the framework on both the broad groups and specific segments. The analysis showed that the framework was able to distinguish varying tiers of healthcare utilization charges as well as relative risk of death, in the following year. The practical tiering of population segments with the framework can thus be directly applied for use in program or policy design, evaluation, and cost-effectiveness analyses.

## Limitations

The healthcare utilization data used primarily covered public healthcare settings and did not include most private healthcare settings. There was a lack of complete data from private acute hospitals and some private primary care, and from community hospitals, nursing homes, hospices, and other intermediate and long-term care (ILTC) services, which were more commonly utilized by individuals with cancer, with transitional care needs, who are frail, and in the last year of their lives. As such, (1) individuals who only utilized private healthcare with no contact with the public healthcare system, and (2) individuals with only utilization of ILTC services would not be accurately assigned to the correct segments. The healthcare utilization and charges are also an underestimation.

Another limitation is that the HealthSCOPES framework was developed specifically with Singapore's setting and healthcare system in mind. The overall conceptual framework and the general classification approach could be translated to and tweaked for another nation's population. However, the exact classification criteria and detailed analyses of healthcare utilization charges might not be applicable unless similar data exist and can be linked at a national level.

The outcome measure of focus in this paper was on healthcare utilization charges and did not include other societal and personal costs incurred. This was again limited by data availability, with societal and personal costs incurred difficult and thus not routinely collected systematically on a national level. Future work for the HealthSCOPES framework would be to incorporate indicators of interests specific to each segment that measures short- and long-term outcomes, as well as process indicators of any nation-wide programs. The long-term stability of the health segments over a longer-term duration, which was not assessed here due to limited data in subsequent years, could also be explored in future work.

## Conclusion

The HealthSCOPES framework developed was used to systematically analyze existing data and could also allow for integration of new data to continue tracking of the health of population. The framework was designed to cover the entire population in Singapore while maintaining relevance to the healthcare needs and setting. It was found that segments comprising individuals with cancer, with transitional care needs, and in the last year of their lives had the highest mean per resident healthcare charges. The use of such a health segmentation framework and analysis allows for better understanding and monitoring of health profiles in the population, which can support data-driven resource allocation based on population health needs and is an anchoring pathway for sustainable and equitable care delivery.

## Acknowledgments

We would like to thank Deanette Pang and Zheng Yi Lau for supporting the access to the databases. We would also like to thank John Wong, Noreen Chan, Jeremy Lin, Milawaty Nurjono, Keith Chong, Teresa Quek, Xu Weiming, and Jaslyn Wong for their feedback during the development of the framework.

## Author Contributions

**Conceptualization:** Ian Yi Han Ang, Kyle Xin Quan Tan.

**Data curation:** Kelvin Bryan Tan.

**Formal analysis:** Nabilah Rahman, Shing Hei Wong, Sheryl Hui-Xian Ng, Ke Xin Eh.

**Methodology:** Ian Yi Han Ang, Nabilah Rahman, Shing Hei Wong, Sheryl Hui-Xian Ng, Kyle Xin Quan Tan, Ke Xin Eh.

**Project administration:** Ian Yi Han Ang, Andrea Su En Lim.

**Resources:** Zheng Jye Ling, Kelvin Bryan Tan.

**Supervision:** Kyle Xin Quan Tan, Zheng Jye Ling, Kelvin Bryan Tan, Sue Anne Toh.

**Validation:** Nabilah Rahman.

**Visualization:** Shing Hei Wong.

**Writing – original draft:** Ian Yi Han Ang, Nabilah Rahman, Shing Hei Wong, Kyle Xin Quan Tan, Andrea Su En Lim.

**Writing – review & editing:** Ian Yi Han Ang, Nabilah Rahman, Sheryl Hui-Xian Ng, Ke Xin Eh, Andrea Su En Lim, Sue Anne Toh.

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
