## [Decision Letter · Decision Letter 0]

19 Aug 2024

PONE-D-24-23865Development and validation of the Health Segment Classification of Population Encompassed within Singapore (HealthSCOPES) frameworkPLOS ONE

Dear Dr. Ang,

Thank you for submitting your manuscript to PLOS ONE. After careful consideration, we feel that it has merit but does not fully meet PLOS ONE’s publication criteria as it currently stands. Therefore, we invite you to submit a revised version of the manuscript that addresses the points raised during the review process.

We look forward to receiving your revised manuscript.

Kind regards,

Apurva kumar Pandya, PhD

Academic Editor

PLOS ONE

Journal Requirements:

Additional Editor Comments:

Dear authors, first of all, let me congratulate authors for wonderful work. We received feedback from reviewers and suggest you to make minor revision to consider for publication.

Reviewers' comments:

Reviewer's Responses to Questions

**Comments to the Author**

1. Is the manuscript technically sound, and do the data support the conclusions?

Reviewer #1: Yes

Reviewer #2: Yes

2. Has the statistical analysis been performed appropriately and rigorously? 

Reviewer #1: I Don't Know

Reviewer #2: Yes

3. Have the authors made all data underlying the findings in their manuscript fully available?

Reviewer #1: No

Reviewer #2: Yes

4. Is the manuscript presented in an intelligible fashion and written in standard English?

Reviewer #1: Yes

Reviewer #2: Yes

5. Review Comments to the Author

Reviewer #1: This manuscript presents an in-depth analysis of the development and validation of the Health Segment Classification of Population Encompassed within Singapore (HealthSCOPES) framework.

I have several questions as below

1) Extent and Management of Missing Data:

The manuscript does not explicitly mention the extent of missing data encountered during the study. Could you provide detailed information on the amount of data that was missing? Additionally, what strategies or methodologies were employed to handle the missing data? Understanding the approach to missing data is critical for evaluating the robustness and reliability of the study's findings.

2) Stability of Health Segments:

It is mentioned that the health segments should be stable over time. Can you confirm if this aspect of stability was directly addressed and analyzed within the manuscript? If so, please provide insights into the methods used to assess the stability of the health segments and the results obtained from this analysis.

3) Consultation with Local Clinical Experts:

The development of the HealthSCOPES framework included consultations with local clinical experts. Please elaborate on the number of clinical experts consulted during this process. Additionally, was a consensus reached among these experts regarding the classification criteria and the overall framework? Details on the consultation process and the extent of expert agreement would provide valuable context for the framework's validation.

Explanation of the CHAS Scheme:

4) Given that the audience of the journal may not be familiar with the healthcare system in Singapore, particularly the Community Health Assist Scheme (CHAS), a brief explanation of the scheme and its color coding (e.g., CHAS Blue, CHAS Orange) would be highly beneficial. This context will help readers understand the socio-economic stratifications and their implications on healthcare access and utilization in Singapore.

5) Exclusion of Medifund Status:

The manuscript did not mention whether Medifund status was used in the analysis. Could you provide the rationale behind the exclusion of Medifund status? Understanding the reasons for omitting this financial assistance scheme will clarify any potential biases or limitations in the socio-economic aspects of the health segmentation.

6) Obesity Classification Standards:

Lastly, could you specify whether the classification of obesity in the study follows the World Health Organization (WHO) standards or the Asian Body Mass Index (BMI) criteria for high risk? Given the differences in BMI thresholds for defining obesity in various populations, this clarification is essential for interpreting the findings related to obesity and its associated health risks.

Reviewer #2: The reviewed article seems to me to be very appropriate and its conclusions are based on the findings and results obtained. The methodology is coherent and in accordance with the proposed objectives, as are the conclusions obtained.

It seems to me a good article that contributes and sheds light on patient classification from a multi-criteria perspective, very much to be taken into account by healthcare decision-makers and political managers.

6. PLOS authors have the option to publish the peer review history of their article (what does this mean?). If published, this will include your full peer review and any attached files.

Reviewer #1: **Yes: **Ang Yee Gary

Reviewer #2: **Yes: **AJ Garcia-Ruiz. Del Pharmacology. Health & Economics Research. School of Medicine. University of Málaga (Spain)

---

## [Author Response · Author response to Decision Letter 0]

2 Oct 2024

We thank the editor and reviewers for their comments. Please find the responses to the comments below.

Response: We have reviewed and ensured the style and file naming meets the requirements. 

2. We note that you have indicated that there are restrictions to data sharing for this study. For studies involving human research participant data or other sensitive data, we encourage authors to share de-identified or anonymized data. However, when data cannot be publicly shared for ethical reasons, we allow authors to make their data sets available upon request. 

For information on unacceptable data access restrictions, please see http://journals.plos.org/plosone/s/data-availability#loc-unacceptable-data-access-restrictions. 

Response: We have updated the Data Availability statement in the submission to more clearly state that: “Data cannot be shared publicly; there are legal restrictions on sharing the de-identified data because they are national administrative data consolidated and owned by the Government of Singapore. Access to the data can be requested through the TRUST Platform with approvals from the Data Access Committee at https://trustplatform.sg/”

Response: We have reviewed our reference list and it is complete and correct.

Additional Editor Comments:

Dear authors, first of all, let me congratulate authors for wonderful work. We received feedback from reviewers and suggest you to make minor revision to consider for publication.

Response: Thank you.

Reviewers' comments:

Reviewer #1: This manuscript presents an in-depth analysis of the development and validation of the Health Segment Classification of Population Encompassed within Singapore (HealthSCOPES) framework.

I have several questions as below

1) Extent and Management of Missing Data:

The manuscript does not explicitly mention the extent of missing data encountered during the study. Could you provide detailed information on the amount of data that was missing? Additionally, what strategies or methodologies were employed to handle the missing data? Understanding the approach to missing data is critical for evaluating the robustness and reliability of the study's findings.

Response: We have added a flowchart to illustrate how the 2016 segmentation population was created. From the master list of individuals, we excluded those without race information, who were non-residents, aged >110 in 2016, with no valid residential postal code and who died in 2015 or earlier. 

For the summary statistics on the segment demographics (Table 2), summary statistics on healthcare utilization (Fig 3) and validation analysis (Table 3, Fig 4 and Table 4), there was no further missing data. For each of the aforementioned results, we have since indicated in the manuscript that data was complete.

Rather than missing data, the limitation of the study’s findings is lack of additional types of data to apply some of the segmentation criteria. This limitation can be found in the discussion section.

2) Stability of Health Segments:

It is mentioned that the health segments should be stable over time. Can you confirm if this aspect of stability was directly addressed and analyzed within the manuscript? If so, please provide insights into the methods used to assess the stability of the health segments and the results obtained from this analysis.

Response: Stability aspect of the health segments is not addressed and analyzed in this manuscript. To assess the stability aspect of the manuscript, health segments of multiple years need to be available. At the point of analysis, not all of the equivalent dataset segments used to segment 2016 population were ready and available for the following years. Hence, we were not able to assess stability well without attributing any differences to the lack of equivalent data in applying the segmentation criteria for subsequent years. This point is more clearly stated in the Limitations section, with suggestions for future work to explore the multi-year stability. 

3) Consultation with Local Clinical Experts:

The development of the HealthSCOPES framework included consultations with local clinical experts. Please elaborate on the number of clinical experts consulted during this process. Additionally, was a consensus reached among these experts regarding the classification criteria and the overall framework? Details on the consultation process and the extent of expert agreement would provide valuable context for the framework's validation.

Response: Five clinical experts were consulted during the development of the framework. They were at the consultant level and above, and covered specialties of oncology, palliative care, pediatrics, endocrinology, and preventive medicine. The consultations were to first ensure the framework facilitated development of practical care goals, actionable care programs, and measurable outcomes for segments relevant to each clinical expert’s specialty before the quantitative data portion. As such, there were no procedures taken for the overall framework and the classification criteria to reach consensus amongst the clinical experts. We have added these details to the manuscript to clarify the process taken for the framework development.

Explanation of the CHAS Scheme:

4) Given that the audience of the journal may not be familiar with the healthcare system in Singapore, particularly the Community Health Assist Scheme (CHAS), a brief explanation of the scheme and its color coding (e.g., CHAS Blue, CHAS Orange) would be highly beneficial. This context will help readers understand the socio-economic stratifications and their implications on healthcare access and utilization in Singapore.

Response: We have added some description on CHAS scheme in the “Database” section. It now reads:

“The datasets contained healthcare utilization data submitted through the Central Claims Processing System and data submitted for subvention purposes from (i) public sector hospitals for inpatient episodes, emergency department (ED) attendances, and specialist outpatient clinic (SOC) attendances, (ii) public sector primary care centers (polyclinics), and (iii) private sector Community Health Assist Scheme (CHAS) clinics. CHAS is a scheme introduced in 2014 aimed to make primary healthcare more accessible and affordable to lower- to middle-income Singaporean households by providing higher subsidies to these households. Before November 2019, CHAS benefits are tiered depending on household income with CHAS Blue cardholder receiving higher subsidies than CHAS Orange cardholder. From November 2019, CHAS Green was introduced for Singaporean with chronic conditions who do not qualify for CHAS Blue and Orange.”

5) Exclusion of Medifund Status:

The manuscript did not mention whether Medifund status was used in the analysis. Could you provide the rationale behind the exclusion of Medifund status? Understanding the reasons for omitting this financial assistance scheme will clarify any potential biases or limitations in the socio-economic aspects of the health segmentation.

Response: Medifund status was excluded due to repetition with the socio-economic status as capture in the CHAS status. Additionally, Medifund serves as a safety net only triggered in the event of an exorbitant medical bill, and so might not be adequately capturing socio-economic status if not such medical episodes exist. 

6) Obesity Classification Standards:

Lastly, could you specify whether the classification of obesity in the study follows the World Health Organization (WHO) standards or the Asian Body Mass Index (BMI) criteria for high risk? Given the differences in BMI thresholds for defining obesity in various populations, this clarification is essential for interpreting the findings related to obesity and its associated health risks.

Response: Yes, the classification of obesity was set as having body mass index 27.5kg/m2 or higher, the criteria for Asian populations. This is now clearly stated in the manuscript.

Reviewer #2: The reviewed article seems to me to be very appropriate and its conclusions are based on the findings and results obtained. The methodology is coherent and in accordance with the proposed objectives, as are the conclusions obtained.

It seems to me a good article that contributes and sheds light on patient classification from a multi-criteria perspective, very much to be taken into account by healthcare decision-makers and political managers.

Response: Thank you.

While revising your submission, please upload your figure files to the Preflight Analysis and Conversion Engine (PACE) digital diagnostic tool, https://pacev2.apexcovantage.com/. PACE helps ensure that figures meet PLOS requirements. To use PACE, you must first register as a user. Registration is free. Then, login and navigate to the UPLOAD tab, where you will find detailed instructions on how to use the tool. 

Response: The figure files have now been processed through PACE and will be reuploaded.

---

## [Decision Letter · Decision Letter 1]

20 Dec 2024

Development and validation of the Health Segment Classification of Population Encompassed within Singapore (HealthSCOPES) framework

PONE-D-24-23865R1

Dear Dr. Ang,

We’re pleased to inform you that your manuscript has been judged scientifically suitable for publication and will be formally accepted for publication once it meets all outstanding technical requirements.

Kind regards,

Nenad Filipovic

Academic Editor

PLOS ONE

Additional Editor Comments (optional):

Reviewers' comments:

Reviewer's Responses to Questions

**Comments to the Author**

1. If the authors have adequately addressed your comments raised in a previous round of review and you feel that this manuscript is now acceptable for publication, you may indicate that here to bypass the “Comments to the Author” section, enter your conflict of interest statement in the “Confidential to Editor” section, and submit your "Accept" recommendation.

Reviewer #1: All comments have been addressed

Reviewer #2: All comments have been addressed

2. Is the manuscript technically sound, and do the data support the conclusions?

Reviewer #1: Yes

Reviewer #2: Yes

3. Has the statistical analysis been performed appropriately and rigorously? 

Reviewer #1: I Don't Know

Reviewer #2: Yes

4. Have the authors made all data underlying the findings in their manuscript fully available?

Reviewer #1: No

Reviewer #2: Yes

5. Is the manuscript presented in an intelligible fashion and written in standard English?

Reviewer #1: Yes

Reviewer #2: Yes

6. Review Comments to the Author

Reviewer #1: Thank you for making the suggested changes

I am happy to recommend accepting the publication in its current form.

Reviewer #2: (No Response)

7. PLOS authors have the option to publish the peer review history of their article (what does this mean?). If published, this will include your full peer review and any attached files.

Reviewer #1: **Yes: **Dr Ang Yee Gary

Reviewer #2: **Yes: **Antonio J Garcia-Ruiz, MD,MsC,PhD. Dep Farmacólogo & Clinical Therapeutics. Health Economics & Outcomes Research. University of Malaga.

---

## [Editor Report · Acceptance letter]

26 Dec 2024

PONE-D-24-23865R1 

PLOS ONE

Dear Dr. Ang, 

I'm pleased to inform you that your manuscript has been deemed suitable for publication in PLOS ONE. Congratulations! Your manuscript is now being handed over to our production team.

Kind regards, 

on behalf of

Professor Nenad Filipovic 

Academic Editor

PLOS ONE